# Community Stakeholders’ Perceptions on Barriers and Facilitators to Food Security of Families with Children under Three Years before and during COVID-19

**DOI:** 10.3390/ijerph191710642

**Published:** 2022-08-26

**Authors:** Elder Garcia Varela, Jamie Zeldman, Amy R. Mobley

**Affiliations:** Department of Health Education and Behavior, College of Health and Human Performance, University of Florida, Gainesville, FL 32611, USA

**Keywords:** food security, nutrition, early childhood, COVID-19, community stakeholders, PRECEDE–PROCEED model

## Abstract

Children living in food-insecure households have poorer overall health than children in food-secure households. While U.S. nutrition assistance programs provide resources, these cannot consistently offer age-appropriate nutritional foods for young children. This study aimed to determine community stakeholders’ perceptions of the barriers and facilitators to obtaining adequate, high-quality, and age-appropriate foods for children ages 0–3 in Florida before and during COVID-19. Community stakeholders (*n* = 32) participated in a 60 min interview via Zoom using a semi-structured script based on the PRECEDE component of the PRECEDE–PROCEED model. Interviews were transcribed verbatim and coded by two researchers using a thematic analysis approach. Stakeholders’ perceptions revealed a lack of awareness surrounding eligibility for assistance programs, a lack of knowledge regarding how to obtain resources and services, and stigma associated with receiving benefits. These remained significant barriers to obtaining healthful foods for households with young children before and during COVID-19. Nonetheless, barriers were exacerbated during the pandemic. Unemployment rates rose, intensifying these households’ financial hardships and food insecurity levels. Likewise, stakeholders suggested the need for families to become more aware of federal assistance eligibility requirements and available opportunities via social media and referrals. Identifying risk factors associated with food insecurity can inform future interventions to safeguard young children’s health and well-being.

## 1. Introduction

Food insecurity (i.e., limited or uncertain availability of nutritionally adequate and safe foods, or limited or uncertain ability to acquire acceptable foods in socially acceptable ways) affected 10.5 percent (13.8 million) of all U.S. households and 14.8 percent (5.6 million) of households with children under age 18 at some time during 2020 [1]. Specifically, among U.S. households with children under age 6, 15.3 percent (2.5 million) reported being food insecure [1]. Food insecurity and poor nutrition are closely linked, with poor nutrition impacting many chronic diet-related diseases, including an increased risk of obesity, diabetes, and heart disease. As a result, the Biden–Harris administration has made it a core priority to promote food and nutrition security. Nutrition security refers to “consistent and equitable access to healthy, safe, affordable foods essential to optimal health and well-being.”

On 11 March 2020, the World Health Organization declared the spread of SARS-CoV-2 (COVID-19) a global pandemic [2]. The COVID-19 outbreak created a public health and economic crisis, disrupting the lives of many Americans [3]. The U.S. unemployment rate in April 2020 rose from 10.3 to 14.7 percent, the highest and the largest over-the-month increase since 1948 [4]. While employment in the U.S. began to rebound within a few months, considerable unmet needs remained near the end of 2021. Millions of U.S. households reported having difficulty obtaining nutritionally adequate foods and being able to afford other necessities such as rent [5].

During the first 1000 days of a child’s life, children experience rapid physical, cognitive, developmental, and social growth [6,7,8,9,10,11,12,13]. Nutrition and environmental factors (i.e., access to affordable healthy food options) play essential roles in a child’s development and the prevention of health conditions [14,15,16,17,18,19]. Research suggests that a lack of adequate nutritious food is associated with nutrient deficiencies, creating learning and development problems [6,7,20,21,22,23]. Specifically, micronutrient deficiencies (i.e., iodine, iron, and zinc) have been associated with developmental risk and poor cognitive function (i.e., aggression, anxiety/depression, and reading and attention problems) [6,7,20,21,22,23]. Research also suggests food insecurity contributes to unhealthy and irregular eating patterns and an increased risk of obesity [24,25,26,27].

Existing U.S. nutrition assistance programs (i.e., Supplemental Nutrition Assistance Program (SNAP), Special Supplemental Nutrition Program for Women, Infants and Children (WIC), Child and Adult Care Food Program (CACFP), and Early Head Start (EHS)) provide supplemental food and educational resources, alleviating food insecurity among eligible families with young children [28,29,30,31]. Likewise, emergency food programs (i.e., food pantries, food banks, and soup kitchens) also provide a source of a wide variety of shelf-stable products and fresh foods [32]. In 2020, 6.7 percent of all U.S. households reported using a food pantry, an increase from 4.4 percent in 2019 [1]. Lockdown, stay-at-home, and school closure policies resulting from COVID-19 exacerbated the need for and utilization of emergency food service provisions for children and families [33]. For instance, the unprecedented unemployment rate created long lines at food banks as families struggled to access low-cost foods [34]. At the same time, interruptions in supply chains and lockdowns also prompted some consumers to stockpile shelf-stable groceries, decreasing the availability of affordable foods for these families [33,35].

While several U.S. nutrition assistance programs (i.e., WIC, SNAP, CACFP, etc.) provide supplemental food and educational resources to eligible families, some families are unable to consistently provide high-quality nutritional items that are age-appropriate for their infants and toddlers [36,37,38]. Furthermore, emergency food programs cannot always provide sufficient food to protect families from experiencing food insecurity for more than a few days [29,39]. The COVID-19 pandemic further exposed strengths and weaknesses in existing nutrition-related policies and emergency food programs [40]. For instance, although the federal government quickly increased benefits and reduced barriers to access federal assistance programs (i.e., SNAP, WIC), as well as implemented new programs to provide additional funds to families to improve access to nutritious food, some of these emergency food programs and food assistance policies were discontinued as the COVID-19 crisis abated [40]. Therefore, it remains imperative to explore the needs of individuals in food insecure households with young children in order to support the expansion and innovation of policies and programs that promote and provide the consistent and equitable access, affordability, and utilization of nutritious and age-appropriate foods [40].

The objective of this study was to explore community stakeholders’ (i.e., community and policy outreach providers, early childhood educators, emergency food providers, healthcare providers, and nutrition educators) perceptions of the barriers to and facilitators of food security in households with children under age three in low-income communities before and during COVID-19. Community stakeholders offer a unique perspective regarding community needs, questions, concerns, and individuals’ expectations, behaviors, attitudes, and values [41,42]. Thus, assessing community stakeholders’ perceptions is an appropriate first step to better understand the factors influencing food security in households with infants and toddlers, and ultimately promote evidence-based policies and programs that improve young children’s health and well-being.

## 2. Materials and Methods

This study is an exploratory qualitative study using semi-structured interviews informed by the Predisposing, Reinforcing, and Enabling (PRE) Constructs in the Educational Diagnosis and Evaluation (PRECEDE) component of the PRECEDE–PROCEED model. This model is an ecological approach to health promotion used extensively in previous health promotion interventions in nutrition education and obesity prevention [43]. Predisposing factors refer to the antecedents to behavior that provide the rationale or motivation for the behavior. Reinforcing factors indicate elements that provide continuing reward or incentive for the persistence or repetition of the behavior. Enabling factors are antecedents to behavioral or environmental change that allow a motivational or environmental policy to be enforced or implemented. This study serves as formative research to guide and inform the design of a community-based intervention to address food security in the target population. The COREQ (COnsolidated criteria for REporting Qualitative research) checklist was used as a guide in reporting the methods and results (Appendix A).

### 2.1. Participants

A purposive sample of key community stakeholders with a vested interest in families with children ages 0 to 3 in Florida were recruited to participate in this study (response rate of 85%). Purposive sampling ensures that the perspectives of identified stakeholders are represented [44]. Community stakeholders included those working in private, non-profit, and government organizations in the fields of community and policy outreach (e.g., research and policy centers), early childhood education (e.g., childcare), emergency food provision (e.g., food pantries), healthcare (e.g., pediatricians) and nutrition education (e.g., WIC, SNAP, EFNEP educators). Exploring the perspectives of different key stakeholder groups ensured that the topic (i.e., food security in households with young children) is not explored through one lens but rather a variety of lenses as these individuals interact directly or indirectly with the caregiver and the child in different capacities.

Participants were recruited via email through existing community connections and electronic announcements. Flyers were sent electronically via email to identified organizations in six counties in the state of Florida (i.e., Alachua, Duval, Lake, Orange, Palm Beach, and Volusia). These counties are in the southeast, central, and northeast districts of the state. The study staff also placed phone calls to follow up with individuals interested in participating. Participants received monetary compensation in the form of a $30 electronic Amazon gift card after completing the interview.

### 2.2. Data Collection

One-time, one-on-one, semi-structured interviews were audio-recorded and conducted via Zoom, lasting approximately 60 to 90 min, between November 2019–August 2021. Interviews were conducted until data saturation was reached. Saturation was guided by the seven parameters identified by Hennink et al. (i.e., study purpose, population, sampling strategy, data quality, type of codes, code book and saturation goal, and focus retrieved from the study) [45]. Before each interview, participants provided a waiver of consent and completed a brief demographic survey (Appendix B) via Qualtrics. Moderator guides informed by recent related reports were used to facilitate participant interviews (Appendix C) [46]. Recordings were transcribed and cross-checked for accuracy.

### 2.3. Data Analysis

Moderators and notetakers were trained in interview methods [47]. Descriptive statistics of the demographic data were summarized using IBM SPSS Statistics 23. Qualitative data were deidentified to remove any personal identifiers before analysis. For this paper, the following questions were selected for analysis: (1) What do you think were the major barriers to food security for local families with infants and toddlers ages 0–3 years prior to COVID-19 (before March 2020)? (2) What do you think were the major barriers to food security for local families with infants and toddlers ages 0–3 years during the COVID-19 response (since March 2020)? (3) What foods are most challenging for parents to obtain for infants and toddlers ages 0–3 years because of COVID-19? (4) What resources were available in the community to address these barriers prior to COVID-19? And (5) What resources are available in the community to address these barriers because of COVID-19?

Two experienced female research team members (EGV and JZ) analyzed the data collected via traditional text analysis. The research team coded the transcripts independently, following an inductive content analysis methodology. After the initial coding of participants’ responses, the research team met to compare similarities and discrepancies in the coding and reach a consensus [48]. An Inter-Rater Reliability (IRR) of 80% agreement between coders on 95% of the codes was established among multiple coders to mitigate interpretative bias and maintain coding consistency [49]. The identification of categories followed the same structure. The PRECEDE component of the PRECEDE–PROCEED model was conceptually used to identify the themes and subthemes. Qualitative crosstabulation analysis compared pre and existing COVID-19 data among participants. To increase the accuracy of predetermined categories and achieve unbiased results, another research team member (ARM) reviewed the findings.

## 3. Results

### 3.1. Demographic Characteristics

Participant characteristics can be found in Table 1. A total of 32 community stakeholders were interviewed with equal representation across key community stakeholder groups (i.e., community and policy outreach providers, early childhood educators, emergency food providers, healthcare providers, and nutrition educators). Most participants were white non-Hispanic females with an average of 8 years of experience working with low-income households with young children.

### 3.2. Results

The following summarizes the community stakeholders’ perceptions of the barriers and facilitators to obtaining adequate, high-quality, and age-appropriate food for children under three before and during COVID-19. Results are organized according to Phase 4: Educational and Ecological Assessment of the PRECEDE–PROCEED model.

#### 3.2.1. Barriers to Food Security for Households with Children under Three Years

##### Predisposing Factors

Overall, community stakeholders suggested the following predisposing factors as the main barriers to food security for households with children under three years before COVID-19: (1) a lack of knowledge about the application process, logistics, and eligibility for federal assistance programs and opportunities; (2) a lack of trust and motivation to utilize federal assistance programs; (3) a lack of knowledge regarding how to obtain resources and services; (4) stress associated with life events; and (5) pride and/or stigma associated with receiving food and other assistance. Stakeholders mentioned that while new barriers emerged due to COVID-19, existing barriers were exacerbated. Participants mentioned the following predisposing factors that created new barriers to food security for households with young children: (1) COVID-19 impacts on diet-related behaviors, sedentary behaviors, and mental health; and (2) CDC guidelines related to COVID-19 safety measures. Table 2 provides quotes from participants regarding the identified predisposing factors.

Regarding COVID-19 impacts on diet-related behaviors, stakeholders described caregivers’ fear of food shortages which prompted them to purchase greater quantities of cheaper foods that were often low in nutritional quality, further impacting existing diet-related behaviors. Similarly, stakeholders also suggested that the fear of exposure to COVID-19 prevented caregivers from accessing required or desired foods. Mental health impacts were noted by participants when they shared how the pandemic made it challenging for caregivers to balance existing responsibilities (i.e., work, childcare, household chores, etc.) with new demands (i.e., home-schooling, workloads, the fear of infection, etc.), especially if they had more than one child. As a result, these caregivers reported increased depressive symptoms affecting food-related parenting practices.

Additionally, participants stated that the caregivers of young children did not want to put their children at risk by going to the supermarket or food pantry. Furthermore, because caregivers faced challenges with access to childcare services and support, they could not leave their children unattended at home and had to bring them wherever they had to go, putting them at a risk of contracting the virus. Additionally, stakeholders stated caregivers expressed concern about their children being potentially exposed to COVID-19 because individuals in their communities were not following the required and recommended CDC safety guidelines and protocols to prevent the spread of the virus.

##### Reinforcing Factors

Regarding reinforcing factors, participants suggested relationships within the home environment created a barrier to food security for these households before COVID-19. For instance, unsolicited advice or pressure from other family members, especially grandparents and family members from older generations, may negatively impact child feeding. One participant mentioned that caregivers living with their parents or grandparents felt pressured to follow the grandparents’ recommendations regarding feeding practices (i.e., giving dairy milk or juice to the child before they are six months old). Access to reliable and supportive networks or systems continued to be challenging during the pandemic. Participants mentioned that the rapid spread of COVID-19 and high infection rates made it challenging for these families to rely on the support of families and friends for childcare and transportation. Some families who do not have a primary mode of transportation rely on family members’ mode of transport to access food at supermarkets and other establishments. Likewise, caregivers were forced to stay home for childcare in lieu of workingbecause they could not have their support system watch over their children with schools and childcare facilities being closed.

##### Enabling Factors

Moreover, community stakeholders identified the following enabling factors as barriers to food security for households with children under three years before COVID-19: (1) a lack of or limited access to transportation; (2) a lack of or limited access to affordable housing; (3) a lack of or limited access and availability to healthy foods; (4) financial insecurity; and (5) a lack of access to healthcare resources/services. In comparison with predisposing and reinforcing factors, community stakeholders identified additional enabling factors resulting from the pandemic. Participants stated that in addition to the barriers above, households with children under three years faced (1) higher unemployment rates; (2) COVID-19 restrictions (i.e., social distancing, self-isolation, shutdowns, curfews, etc.) due to virus exposure; (3) a lack of or limited access to technology or internet and connectivity issues; (4) food distribution pick-up challenges (i.e., long wait times, restrictions, etc.); and (5) a lack of or limited affordable childcare services. For instance, participants indicated caregivers having difficulty obtaining certain foods due to COVID-19 restrictions. Stakeholders mentioned caregivers had problems acquiring baby formula, fresh produce (i.e., fruits and vegetables), protein, and dairy products during the pandemic. Specifically, stakeholders mentioned that food pantries rarely provide baby-friendly foods. One participant noted, “One of the biggest barriers is that we [food pantry] run out a formula in the middle of the month. And we are not able to get them until a week later, two weeks later, maybe the following month. So not having enough resources to meet the demand of the baby because that’s the only baby food we provide.” Nevertheless, participants stated that information provided to caregivers through nutrition education programs and the rapid changes in legislation to increase SNAP and WIC benefit utilization made it easier for caregivers to obtain these foods during this time.

#### 3.2.2. Facilitators Available in the Community to Address Barriers to Food Security for Households with Children under Three Years

When asked about the services and resources available to address barriers to food security before COVID-19, community stakeholders mentioned several resources, including (1) information and referral services; (2) health promotion programs; (3) nutrition programs; (4) volunteer and intergenerational programs, and (5) housing programs. Information and referral services, as well as nutrition programs such as federally sponsored programs (i.e., SNAP, WIC, etc.), food banks, food pantries, and community gardens, were the most mentioned among stakeholders.

While participants mentioned most of the programs, services, and resources available before COVID-19 were still offered during the pandemic, they were economically impacted by COVID-19 response efforts (i.e., lockdowns and social distancing measures). Community-based organizations faced dwindling resources despite the increased needs of individuals in communities. Nevertheless, stakeholders stated existing and new programs and organizations rapidly adapted and adopted new strategies to provide programming and resources to those in need, including virtual support and socially distanced measures. Volunteer and intergenerational programs and nutrition programs remained the most accessible resources to households with young children. Table 3 summarizes the PRE factors influencing the facilitators to obtain adequate, high-quality, and age-appropriate food for children under three in low-income communities before and during COVID-19.

## 4. Discussion

This paper explored community stakeholders’ perceptions of the barriers and facilitators to food security in households with children under three before and during the COVID-19 pandemic. Exploring the perceptions of various stakeholders allowed for a better understanding of the factors contributing to food security in households with young children through diverse lenses, which allows for a more comprehensive picture of the ongoing challenges preventing these families from accessing adequate, high-quality, age-appropriate food for their children.

While stakeholders suggested barriers to food security (e.g., stigma associated with welfare, a lack of knowledge regarding how to obtain resources and services, a lack of awareness/eligibility for supplemental assistance, and a lack of transportation) remained the same during the COVID-19 response, new challenges emerged (e.g., COVID-19 implications on diet-related behaviors, sedentary behaviors, and mental health; higher rates of unemployment, and COVID-19 safety protocols) due to possible virus exposure. Understanding the barriers that emerged during the COVID-19 pandemic highlights the challenges that often arise during times of crisis (e.g., global pandemics, natural disasters, financial crises). It also encourages conversation and policy action toward future changes in social, economic, and environmental systems.

Recent studies have found similar findings suggesting individuals consumed high rates of diets high in saturated fats, sugars, and refined carbohydrates, known to be contributors to obesity and type 2 diabetes [50,51,52]. Moreover, other studies have also found that the COVID-19 outbreak caused a significant rise in food prices (e.g., fruits and vegetables, meats, dairy products, etc.) related to lockdown restrictions accompanied by panic buying, as well as supply chain disruptions [53]. For instance, safety protocols and measures brought unique challenges to the food industry and its supply chain when isolation and quarantine orders increased consumption of and demand for non-perishable shelf-stable foods (i.e., canned foods, grains, and starchy foods) [35,54,55]. Similarly, a qualitative study found that the fear of food shortages and starvation during the onset of COVID-19 was a motivator for parents to stock up on food with a long shelf-life [56]. Unfortunately, these families stated that purchasing more energy-dense foods is convenient, more filling, and less expensive [56].

While previous research has also identified unemployment, a lack of transportation, and the inability to prioritize food purchasing (i.e., paying bills takes priority over buying food) as contributing factors to food insecurity in adults, our findings provide insight into the unique barriers that families with young children face regardless of the impact of COVID-19 [8,13,57]. For instance, participants suggested the supply of the foods available in emergency food assistance programs is scarce, specifically in reference to food items for infants and toddlers. Community stakeholders also mentioned that while some food pantries provide baby formula, these are often only available for a couple of weeks every few months. Participants stated this is often due to low demand from community members, suggesting there is an assumption that baby-friendly foods are not staple items in emergency food assistance programs. For example, one participant mentioned “some families that had physically come when our food pantry was open, they have requested and asked for specific formulas and baby foods. All we had was ten cans of formula and [they] were gone within a week. So, it’s beginning to become something more requested, and not easily available.”

Furthermore, despite shifts in emergency food assistance and federal nutrition assistance programs to increase food access and buffer against the loss of household income during the pandemic [58], stakeholders reported that families with children under age three experienced fear resulting from virus exposure and stigma associated with receiving benefits along with technology demands. A lack of transportation also contributed to lower participation rates among households with young children. Consequently, some households could not access or purchase adequate and age-appropriate foods, including fruits and vegetables and baby formula.

Community stakeholders also expressed caregivers’ lack of knowledge and awareness about programs and resources addressing food security. As with previous research, findings from this study suggested this is due to the lack of community outreach opportunities and logistics associated with applying for and receiving federal, state, and local supplemental assistance [59,60]. Moreover, participants also stated the stigma associated with applying for or receiving additional assistance hindered caregivers’ ability to provide for their children, especially during the pandemic. Research has found that shame and anxiety can intensify the stigmatization of participation in food assistance programs and the acceptance of charitable foods [61,62]. Thus, even if resources and services were available in these communities, there is a low participation rate due to a lack of knowledge and the stigma associated with receiving benefits.

Furthermore, community stakeholders mentioned that individuals who lost employment yet whose household income was above the income eligibility threshold could not receive financial or food assistance benefits and support. The existing literature supports this finding, suggesting eligibility restrictions and enrollment barriers intensify the challenges for these households to access food and other resources [63,64]. Overall, the results from this study provide a foundation to explore policy recommendations and opportunities to reduce social and economic disparities, especially in vulnerable populations. As such, there is a need for legislation that reduces financial hardship through social assistance to improve the health and quality of life of households with young children.

### Limitations

While this study provides a unique insight into the barriers and opportunities related to food security in households with children under three, there are potential study limitations concerning the generality of the findings. First, study participants were part of a purposive sample of community stakeholders in one U.S. state (i.e., Florida). Thus, participants’ responses may not represent the perspectives of all types of stakeholders working with low-income families with children under three in Florida or beyond Florida. However, rigorous data collection techniques were utilized to diminish any potential bias. Additionally, given qualitative data were collected before, during the onset, and in the midst of the COVID-19 pandemic, participants’ responses may represent the political, economic, and social climate these communities were facing in response to the COVID-19 emergency response. It is important to note that depending on when the data were collected, participants may not have been able to recall all the barriers present or facilitators available before the COVID-19 pandemic. Likewise, given participants were sharing the barriers of individuals they served in their current positions, they may not have known all the barriers contributing to food security in this population, adding potential bias to the results. Furthermore, given that food insecurity affects all members of the household, it was challenging for stakeholders to identify barriers and facilitations impacting only children under the age of three. Thus, future research should explore the perceptions of caregivers of children under three years. Lastly, questions and probes utilized in this study could have potentially influenced the personal opinions, perspectives, and experiences of participants.

## 5. Conclusions

Food insecurity is a public health issue affecting millions of households with children in the U.S. [1] The COVID-19 pandemic further exacerbated the state of food security for these households creating a time of deep economic hardship and physical and mental distress for families who were already struggling to make ends meet [34]. Research suggests food insecurity has a detrimental effect on children’s health, growth, and development [13]. Thus, understanding the factors that contribute to food security in these households is pivotal for addressing the needs of this population and informing the development of programs and policy solutions to improve food access and diet quality for young children.

This paper explored community stakeholders’ perceptions of the barriers and facilitators to food insecurity experienced in households with children ages 0–3 in Florida before and during COVID-19. Stakeholders indicated that the families they work with identified stigma associated with applying for or receiving nutrition assistance benefits, a lack of knowledge on obtaining community resources and services, and a lack of transportation as barriers to food security prior to the COVID-19 pandemic. Further, stakeholders identified new challenges for families in accessing healthy and age-appropriate foods for their young children, including the fear of COVID-19 exposure when accessing food or benefits, and concern for higher unemployment rates and resulting negative impact on family income. Stakeholders also reflected on the opportunities available to these families to mitigate the barriers identified. They suggested both government and community organizations create innovative strategies (e.g., the virtual delivery of services and resources) to maximize the opportunities available to these families to facilitate the access to and availability of nutritious and adequate foods for their young children.

Despite existing policies and programs, significant gaps remain in achieving optimal food and nutrition security for families with children under three. Age-appropriate foods and additional resources are still needed and must be provided in non-stigmatized ways for food-insecure families with young children ages three and under, especially now in response to the COVID-19 pandemic. Food assistance policies and programs to achieve nutrition security should ensure optimal utilization by improving coordination across programs and providing equitable and non-stigmatized access to services and resources to improve access to healthy and affordable foods. Future research should assess existing policies and programmatic efforts to provide the equitable and stable availability, accessibility, affordability, and utilization of food with sufficient nutritional quality for young children.

## Figures and Tables

**Table 1 ijerph-19-10642-t001:** Community Stakeholders’ Demographic Characteristics (*n* = 32).

Characteristic	Mean ± SD	Percentage
Age (years)	41.87 ± 11.416	
<30		19.4
30–40		25.8
41–50		32.3
>50		21.9
Years of Work Experience	8.09 ± 7.806	
<10		65.6
10–20		25
>20		9.4
Gender		
Male		6.3
Female		93.8
Race		
Asian		6.3
Black or African American		21.9
White		62.5
Other		9.4
Ethnicity		
Hispanic/Latino(a)		9.4
Not Hispanic/Latino(a)		90.6
Education		
High school graduate or GED ^1^		0
Some college or technical school		12.5
Associate’s degree		3.1
Bachelor’s degree or higher		84.4

^1^ GED: Tests of General Educational Development.

**Table 2 ijerph-19-10642-t002:** Predisposing, Reinforcing, and Enabling (PRE) factors influencing the barriers to obtaining adequate, high-quality, and age-appropriate food for children under three in low-income communities before and during COVID-19.

Barriers to Food Security for Households with Children under Three Years				
PRE Factors	Before COVID-19	Quotes	During COVID-19	Quotes
Predisposing	Lack of knowledge about the application process, logistics, and eligibility for federal assistance	“A lot of [our clients] don’t know that [food assistance programs] exist or that they are eligible. Some don’t want it, or some don’t know how to fill it out. Some don’t speak English and are afraid of providing personal information.”	Responses remained the same. COVID-19 only exacerbated existing barriers	“It’s a very high-stress time for all families. For folks that had never experienced it [food insecurity] before and [COVID-19] exacerbatedfor those that were experiencing it prior. Transportation is probably an even bigger concern now, and people are less likely to want to carpool… families are concerned with social distancing policies.”
	Lack of trust and motivation to utilize federal assistance programs	“I think there… there can be a lack of will to [be] enrolled based on concern about… government being involved in what’s happening with your family and trust on that in that front.”	COVID-19 impact on diet-related behaviors, sedentary behaviors, and mental health	“So not having access to the grocery stores… Kids are mainly eating junk food and not being able to get their WIC foods in time.”
	Lack of knowledge of how to obtain resources and services	“Families may not know what services are available. [Some families] may know about the services, but they have the misconception that it is only for extremely poor or only single moms.”	Lack of compliance with CDC guidelines related to COVID-19 safety measures	“I’ve had a lot of people showing up [in person] more recently that were sick, and you know they don’t really believe in COVID, or for whatever reason, they are still going about things like normal, no fear related to deaths related to COVID.”
	Stress associated with life events	“We have mental health and substance abuse issues in this community. It’s difficult for people to seek treatment and stay in treatment, especially if you have other stressors in your life like many people do that are in poverty.”		
	Pride and/or stigma associated with receiving food and other assistance	“When you talk to families with young children participating in programs… where they have to physically go to pick up food…there is the concern about who’s going to see me.”		
Reinforcing	Home environment (relationships, feeding practices, etc.)	“It can be that a child has multiple different caretakers. They may move between different family members, and I think that’s difficult.”	Responses remained the same. COVID-19 only exacerbated the existing barrier	“Clients didn’t have family members to lean on because they probably couldn’t see those family members or were afraid to do so. The family network that would’ve helped before… that might have been a new barrier... the inability to access their family members to help.”
Enabling	Lack of or limited access to transportation	“Transportation is probably the biggest barrier to families. Families with young children have trouble accessing healthy foods and nutrition education.”	Responses remained the same. COVID-19 only exacerbated the existing barrier	“Now the issue more so is transportation to get to the pantry or just, you know, having to be creative in finding ways to get clients food while we’re dealing with COVID.”
	Lack of or limited access to affordable housing	“We’re a very expensive county, so affordable housing is a challenge. Money tends to go towards high housing and leaving less money for purchasing a food. That’s really what a lot of our families struggle with.”	Higher unemployment rates	“Going into COVID, once you see the high level of unemployment or just people that could potentially be at risk and choosing to stay at home and not really be out and about, or because they have to take care of their kids. There were some difficult situations that families did go through where it was like, okay, I have this much money, but rent is due…I have x amount of kids to feed; what do I do?”
	Lack of or limited access and availability to healthy foods	“We see a big problem over there with them having issues to access to, you know, food, and especially nourishing food because it’s a food desert.”	COVID-19 restrictions (i.e., social distancing, shutdowns, etc.) due to virus exposure	“Families couldn’t work, you know, because of COVID, whether it was because the company shut down… had restrictions or because they had COVID themselves.”
	Financial insecurity	“The economic state of our state, finding viable work at a living wage has always been a problem. There are disproportionate number of low-income families that may be middle class and are struggling because of their lack of work or lack of resources.”	Lack of or limited access to technology or internet and connectivity issues	“Internet is probably a newer barrier because before COVID-19 it was more the transportation issue. If you could get to a library to use their internet and you could go to the Department of Health for services, now you have to schedule online, or you have to be able to do these things on your own.”
	Lack of access to healthcare resources/services	“We definitely have issues with our families with healthcare and not having adequate health care and having to shift a lot of money towards that.”	Food distribution pick-up challenges (i.e., long time, restrictions, etc.)	“I know that families can’t afford food at the stores, and we’ve tried ramping our resources to provide to them, but they can only come once a month. You know, I wish they could come more often.”
			Lack of or limited affordable childcare services.	“Childcare is a big issue just in general. The pandemic certainly made it worse. Moms ended up leaving jobs because they didn’t really have any other good childcare options for their kids… I see families struggling… basically the job that is available barely pays enough to then also pay for childcare. There’s not a lot of affordable childcare options so then a lot of moms end up not working, while their kids are little because if they did, they don’t even end up taking that much home from that job because of how much the childcare costs.”

**Table 3 ijerph-19-10642-t003:** Predisposing, Reinforcing, and Enabling (PRE) factors influencing the facilitators to obtain adequate, high-quality, and age-appropriate food for children under three in low-income communities before and during COVID-19.

Facilitators Addressing Barriers to Food Security for Households with Children under Three Years				
PRE Factors	Before COVID-19	Quotes	During COVID-19	Quotes
Predisposing	Information and referral services	“We have a 411 network that is trying to be the hub of referral services. They do try their best to refer you to different services with programs that have been registered within the last four months.”	Responses remained the same.	“Everything is virtual. You can upload everything in their [client’s] healthcare district. We are in the beginning stages of a software, but we can receive referrals and send referrals and know who has what from medical to any other tangible assistance based on the needs of our families.”
Reinforcing	Volunteer and intergenerational programs	“I know there is a food pantry, and it’s closely linked to their community garden there to help ameliorate hunger. There are volunteer opportunities.”	Responses remained the same.	“Food pantries definitely increased in size; organizations have had a harder time producing gift baskets… but they’ve been doing a lot more volunteering...”
Enabling	Nutrition programs	“We’re always making sure that we’re referring them to our local WIC program because they have the best services for them. And then we’re able to get them to food banks that provide services local within their area.”	Responses remained the same.	“They are still providing food and assistance where they need it, but I know the one in our county…they’ve kind of changed their schedule a little bit… people kind of have to make appointments and stuff just because of COVID where before they could come in, just walk in….”
	Health promotion programs	“We have prenatal programs that can kind of help fill the gap, that financial gap where they might be able to save some money and get free diapers and wipes through our community partners.”	Transportation programs	“There is more public transportation up here for sure. So, if you were struggling with the issue of being able to get to appointments and do things…I feel like that’s something that people might have more access to appear as well as things [resources] just not being as like spread out… Better walkability too.”
	Housing programs	“The housing authority offers low-income families a home. We also have Catholic charities who help with mortgage or rent.”	Income support programs	“The county has funding to assist with rental, food, utilities, and now mortgage assistance. They have that online and new software they’re using…”

## Data Availability

Not applicable.

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
