# Peer review of "Community Stakeholders’ Perceptions on Barriers and Facilitators to Food Security of Families with Children under Three Years before and during COVID-19"

_ijerph, 2022, doi:10.3390/ijerph191710642_

Round 1

Reviewer 1 Report

This paper describes the results from a series of interviews with community stakeholders about food insecurity before and during the pandemic. The emphasis on 0-3 year olds and age appropriate foods was confusing to me as a reader because the themes and results do not seem to reflect this objective and instead are related to food security in general. I have included some specific suggestions for each section below. 

Intro

·         Consider shifting some of this framing from food to nutrition insecurity given USDA’s recent shifts

·         On lines 38-43 it may be good to clarify this is in the context of the united states

·         Line 47: which health conditions – would be good to be more specific

·         Micronutrient deficiencies are rare in the US context- would consider refocusing on some of the other harmful effects of food/nutrition insecurity in early childhood

·         Lines 61-63, good point and it would be good to state explicitly why these exacerbated the need for emergency food programs

·         Lines 64-67 need more explanation given this is the motivation for the whole paper – which programs specifically are you referring to? How specifically are they not offering high-quality nutritious, age-appropriate items? The WIC program is designed specifically to do this, although it provides foods in supplemental amounts. The statement that these programs often cannot provide food to protect families for more than a few days is an overstatement and doesn’t seem to reflect the evidence cited or the design of programs such as SNAP and WIC (which were both augmented during COVID to have higher benefit amounts)

·         The introduction should also describe COVID-related changes to federal nutrition assistance programs targeted to families with young children

·         It would be good to lay out in the introduction who the stakeholders are

Methods

·         Would be helpful to include the response rate for interviews- how many invited stakeholders did not participate? How were they different from/similar to those that did participate? How might this bias your results?

·         More information is needed on the types of stakeholders contacted and why – what agencies do they represent – why are they qualified for these interviews?

·         How was the data coded? Was a qualitative software package used?

·         Would suggest using the COREQ checklist for more transparent reporting of qualitative methods used

·         More information is needed on the geographic region(s) and demographic characteristics of the regions represented by these stakeholders – what implications does this have for the interpretation of your results?

Results

·         To help the reader, I’d recommend adding subheaders for the Predisposing, Reinforcing, and Enabling factors and adding references to Table 2 in section 3.2.1.

·         There are sections of the results where there is important content, but no clear theme identified (e.g., lines 179-189) – was this info coded under one of the existing themes?

·         The results and themes need more elaboration and contextualization throughout so the reader can better understand the themes- for example- how did stakeholders perceive that ‘lack of compliance’ to CDC guidelines created new barriers to food security? Or how does unsolicited advice (about what?) contribute to food security?

·      Many, of these themes are not specifically related to providing food for young children – I know you cannot control how people respond to interview questions, but perhaps this should be noted as a limitation – this still seems to be a gap in the literature

·  

Discussion

·         Line 223- would change to ‘possible virus exposure’.

·         Line 227-228 consumption and demand for which  goods- how does this relate to food insecurity?

·         The first paragraph in general is a bit confusing and does not link to the overall objective of the paper which is to understand barriers and facilitators to providing adequate, high-quality, age-appropriate food for children 0-3.

·         The discussion section lacks synthesis with literature about pandemic related shifts in emergency food assistance and federal nutrition assistance programs intended to meet the needs of families with young children -e.g., pandemic-related shifts in the WIC food package and WIC administration, issuance of P-EBT etc. etc.

·         Overall the discussion section is not tailored to the objective of the paper which is specific to this need for adequate age-appropriate food for young children before/during the pandemic- it is not clear to me beyond formula shortages at food pantries where there are facilitators or barriers specific to this topic

·         How do you think the fact that most of the stakeholders have identities that differ from those experiencing food insecurity skewed your results? Would note this as a limitation and talk about how this may/likely biased the results compared to interviews with parents/caregivers themselves

·         How do you think that having specific probes about lack of access/transportation and lack of awareness/stigma influenced the themes that emerged from the interviews? This seems like a major limitation and should be noted in the discussion section

·         Stakeholders’ ability to accurately recall before COVID is a major limitation to this before and during COVID comparison

Conclusions- need to include citations throughout this section

Table 1: what do the means for race, gender, ethnicity, and education refer to? These don’t seem to make sense and I think they should be removed and just leave the percentages of the sample with these characteristics.

Author Response

Thank you for your review.  We have attached a point by point response to all comments and suggestions.

Reviewer 2 Report

Comment 1: In my opinion, the title needs to be modified further.

Comment 2: Abstract: The section needs to be better worked out. In my opinion, it lacks sufficient clarity. The content of the methods in the abstract is not sufficiently clear. I would suggest improving it. The result needs to be improved too.

The authors need to answer these questions in the abstract section: Why did you do the study? What did you do?, What did you find?, What did you conclude?

Comment 3: Introduction: this section looks at insufficient relevant information. So, I suggest to the authors in the introduction section to tell the reader about the question your paper addresses, then summarise relevant literature relating to this issue (not a literature review), and then finish by telling your reader what the rest of the article is going to do.

Comment 4: I miss some justification on the fact that the focus of the study is on the community stakeholders.

Comment 5: I would like to see a clearer justification for undertaking the study. The paper's contribution to the literature needs to be better worked out, and the novelty of the paper should be present in a clear way.

Comment 6: Materials and methods: The section needs to be better worked out. In my opinion, it lacks sufficient detail to be able to fully understand it.

Comment 7: Part 2.3 is not clear. I would suggest presenting this part in a clearer way.

Comment 8: I would suggest to the authors use an Econometric model, like a logistic or probit model, which could be able to identify factors that determine food security in the research area before and after COVID-19.

Comment 9: Result: Part 3.2.1 and Part 3.2.2 should be written in a more clear way and should be supported by the finding of the table. However, I didn’t see your finding in the manuscript to support your research question. So, the authors should clearly present their findings in the result section.

Comment 10: Readability needs to be improved, especially at certain points. I would suggest writing in a more “concise” manner at certain points too.

Author Response

Thank you for your review.  We have attached a point-by-point response to all comments and suggestions.

Reviewer 3 Report

2.1 participants - provide a bit more detail on types of organizations recruited.

2.3 data analysis - unclear if coding categories were established prior to coding or if inductive coding was used to develop codes emerging from data.  Clarify

3.1 demographics - I m not sure who are community and policy outreach providers

Table 1 - lists means that are not possible - gender, race, ethnicity etc 

Line 162 - Regarding reinforcing factors, participants suggested relationships within the home 162 environment created a barrier to food security for these households before COVID-19. For instance, unsolicited advice or pressure from other family members, especially grandparents and family members from older generations.  This line requires further explanation

Line 177 - not sure what long time refers to

Line 237 - They stated this is due to low demand from community members, suggesting there is an assumption that baby-friendly foods are not stapled items in emergency food assistance programs, I  am not interpreting the results in the same way.  Perhaps include quotes that support this statement.

line 255 - add some additional policy /programming recommendations that arise from study findings.

The conclusion should be improved by including specific important results of this study.

Author Response

Thank you for your review.  Please see attached response to all comments.

Reviewer 4 Report

The authors determine community stakeholders’ perceptions of the barriers and facilitators to obtaining adequate, high-quality, and age-appropriate foods for children ages 0-3 in Florida before and during COVID-19. This is an important and relevant subject, but some points are necessary to improve in the manuscript before publications.

Introduction

First, it is not clear why the authors considered only the community stakeholders’ perceptions and not the parents/caregivers. In the introduction section it must be clear the role of these professionals in the USA community assistance programs and the importance of their opinion.

Lines 32-34: please corroborate the proportion values, because it seems there is some confusion or little clarity about the numbers. How is it possible that the proportion for those under 6 (15.3%) be higher than those under 18 (14.8%)? Please review redaction and improve it.

 The author must carefully review the pertinence of the cited references. For example, the references 6-13 related to this declaration: “During the first 1000 days of a child’s life, children experience rapid physical, cognitive, developmental, and social growth [6-13].  Is there necessary all of it? Are there the best to substantiate this declaration?  Justify. This happen in more part of the manuscript.

Material and methods

Line 90: Please provide more details in relation to the selected sample of key community stakeholders. Are they from the same district/regions/municipality? Were there any selection criteria?

Considering the pandemic framework is important to declare what was the recollection  data period/date.

Thank you for providing the “Appendix B: Interview Moderator Guide”. It is necessary to include information related to the process of elaboration and validation of the questions in the correspondent section.   

Results

-       Table 1. there are some mistakes in relation to the variable analysis. What do the average/ SD of gender, race and ethnicity mean? Nothing. it does not make sense. So, it must be removed and only the percentage remain. Remember that categoric variables are not represented by measures of central tendency and dispersion.

-       Please review and correct the categories of the “Years of Experience” variable in table 1. Where are the people with 25 years of experience?

-       What is GED? Please indicate in footnote.

-       I recommend finding another title for the 3.2. section. The manuscript is not divided in qualitative and quantitative results.

-       Table 2 is so long, I recommend divide it in two, being one as “Barriers to food security for households with children under three years” and the other “Facilitators….”.

In conclusion, please indicate clearly the main identified points that direct answer the objective. Most of the declarations are general recommendations.

Finally, an equal title and very similar abstract (probably with preliminary results), is available in Oral Abstracts from the Journal of Nutrition Education and Behavior Volume 53, Number 7S, 2021".  Clarify this situation.

Author Response

(The authors gave the same response as above.)

Round 2

Reviewer 1 Report

Thank you for your thorough responses, I think the manuscript is much improved and do not have any further suggestions. 

Author Response

Thank you for providing the helpful feedback.

Reviewer 3 Report

My comments have been addressed.  Look at line 383 to see if some words are missing.  Also stapled should likely be just staple.

Author Response

Thank you.  We have slightly revised the text on lines 383-390 and corrected the word stapled to staple on line 317.

Reviewer 4 Report

I do think the manuscript has been improved substantially with all the  modifications. As last recommendation, it is necessary to mention in methods section that this study is part of a larger one.

Author Response

Thank you.  We have added a notation on lines 108-110 that this study is formative research for a larger project.